behaviour/cognition/ecology

parasite avoidance, disgust, *Mandrillus sphinx*, *Macaca fascicularis*, sensory aversion

**Author for correspondence:**
Cécile Sarabian
e-mail: sarabiancecile@gmail.com

# Divergent strategies in faeces avoidance between two cercopithecoid primates

Cécile Sarabian[1], Barthélémy Ngoubangoye[2] and Andrew J. J. MacIntosh[1]

[1]Primate Research Institute, Kyoto University, 41-2 Kanrin, Inuyama 484-8506, Japan
[2]Centre de Primatologie, Centre International de Recherches Médicales de Franceville, Franceville BP 769, Gabon

  CS, 0000-0002-2225-8702; BN, 0000-0002-0868-3868

Parasites constitute a major selective pressure which has shaped animal behaviour through evolutionary time. One adaption to parasites consists of recognizing and avoiding substrates or cues that indicate their presence. Among substrates harbouring infectious agents, faeces are known to elicit avoidance behaviour in numerous animal species. However, the function and mechanisms of faeces avoidance in non-human primates has been largely overlooked by scientists. In this study, we used an experimental approach to investigate whether aversion to faeces in a foraging context is mediated by visual and olfactory cues in two cercopithecoid primates: mandrills (*Mandrillus sphinx*) and long-tailed macaques (*Macaca fascicularis*). Visual and olfactory cues of faeces elicited lower food consumption rates in mandrills and higher food manipulation rates in long-tailed macaques. Both results support the infection-avoidance hypothesis and confirm similar tendencies observed in other primate species. More studies are now needed to investigate the divergence of avoidance strategies observed in non-human primates regarding food contamination.

## 1. Introduction

Parasites, along with predators, constitute a major selective pressure which has shaped animal behaviour through evolutionary time [1]. Analogous to the myriad adaptations of prey organisms to predation, hosts have evolved diverse ways of countering parasites; these are collectively known as the ART of parasite handling [2]. Avoidance (A) consists of a set of actions taken by an organism to reduce its chances of becoming infected, i.e. the behavioural immune system [3,4]. Resistance (R) is the ability of an organism to limit its parasite burden with the help of its immune system, which can be both physiological and behavioural in nature (i.e. self-medication [5]; social immunity [6]). Tolerance (T) is the adaptation

of an organism to living with a given parasite by limiting its harmful effects. Among these three strategies, avoidance is the only preventative measure and is probably the most cost-effective as a result [7].

Among substrates known to harbour infectious agents, faeces are known to elicit avoidance behaviour in a wide range of animal species, including insects [8], rodents [9], ungulates [10], marsupials [11] and proboscids [12]. This is for good reason, as a multitude of infectious organisms (e.g. bacteria, viruses, parasitic protozoa and helminths) abound in animal excreta. Other animal species, however, such as pigs, dogs and rabbits exhibit coprophagy to acquire certain digestive enzymes lacking in their diet, obtain nutrients that were unabsorbed in the gastrointestinal tract during the first passage and/or to develop their microbiome [13]. However, it was only very recently that scientists started testing whether faeces or other faecally contaminated substrates elicit avoidance in non-human primates [14–19]. This group of animals has a popularized reputation for being rather liberal when it comes to their disposition towards faeces, owing to the many anecdotal reports of coprophagy, throwing faeces and even 'painting' their enclosure walls or other substrates with faeces [20,21]. Many such observations were made at captive facilities and are regarded as being abnormal or pathological signs of distress [21], but even in nature, chimpanzees (*Pan troglodytes*), bonobos (*P. paniscus*) and gorillas (*Gorilla gorilla*) may reingest undigested seeds from their own faeces [22–25]. Importantly, however, these behaviours usually concern their own waste products, which pose significantly fewer risks of encountering novel parasites.

Recent experiments and observations show that non-human primates do exhibit faeces avoidance in feeding and social contexts, and use different sensory cues to do so. We now know, for example, that the visual cues of faeces seem to play a role in avoiding faecally contaminated food in Japanese macaques (*Macaca fuscata* [18]) and that olfactory cues of faeces elicit drinking aversion in lemurs (*Eulemur coronatus*, *E. mongoz*, *Lemur catta*, *Varecia rubra*, *V. Variegata* [19]). Faecal odour also reduces tendencies to use tools during foraging tasks in bonobos [15]. Lastly, even tactile cues that mimic the consistency of faeces, but without the associated odour, induce hand-withdrawal reflexes and feeding aversion in chimpanzees [14]. In addition to these outright avoidance behaviours, contaminants such as faeces, soil or sand are also known to trigger food manipulation behaviours such as rubbing and washing in capuchins (*Sapajus apella*), macaques (*M. fascicularis*; *M. fuscata*), vervets (*Chlorocebus pygerythrus*) and great apes, including chimpanzees, bonobos, gorillas and orangutans (*Pongo abelii*) [18,26–29]. All of these behaviours are consistent with the predictions of the parasite avoidance theory of disgust [30], which posits that disgust evolved as an adaptive strategy to counter the often-significant costs of infectious disease [31].

Here, we performed two food-choice experiments with two cercopithecoid species, mandrills (*Mandrillus sphinx*) and long-tailed macaques (*Macaca fascicularis*), to test for the avoidance of conspecific faeces through visual and olfactory cues. Mandrills and macaques are known to rely on both visual and olfactory cues in a variety of socio-sexual contexts [32–37]. A recent study also showed that mandrills appear to use olfaction to avoid interacting with conspecifics infected with potentially pathogenic intestinal protozoa that can readily transmit through social contact; they groomed these infected individuals less frequently than non-parasitized individuals in the group [38]. However, it remains unclear whether the behavioural responses differ when confronted with either visual or olfactory cues of faeces, and further whether any species differences in avoidance responses exist. Several species of macaques have been reported to process (wash, rub, roll) food in various contexts of contamination before ingestion [18,26,39–41]. Though studies of mandrills are far fewer, to our knowledge, similar behavioural patterns have not yet been reported. Therefore, in the present study, and according to the infection-avoidance hypothesis—that animals have evolved strategies to detect and avoid potential sources of infection—we predicted that subjects would be more cautious regarding food associated with faecal stimuli compared to control stimuli, manifest as lower probabilities to feed and/or higher tendencies to manipulate food prior to consumption. We expect the latter to be particularly true for macaques, regarding a preferred food resource, given the previous reports noted above. Our results add to the growing literature on infection-avoidance behaviour in non-human primates, adding phylogenetic coverage to address the diverse ways in which animals cope with the risks of parasites and infectious disease.

# 2. Material and methods

## 2.1. Study site and subjects

Test subjects included a subset of individuals from two groups of mandrills comprising approximately 200 individuals and six groups of long-tailed macaques comprising 22 individuals at the 'Centre International de Recherches Médicales de Franceville' (CIRMF) in South-East Gabon. We tested

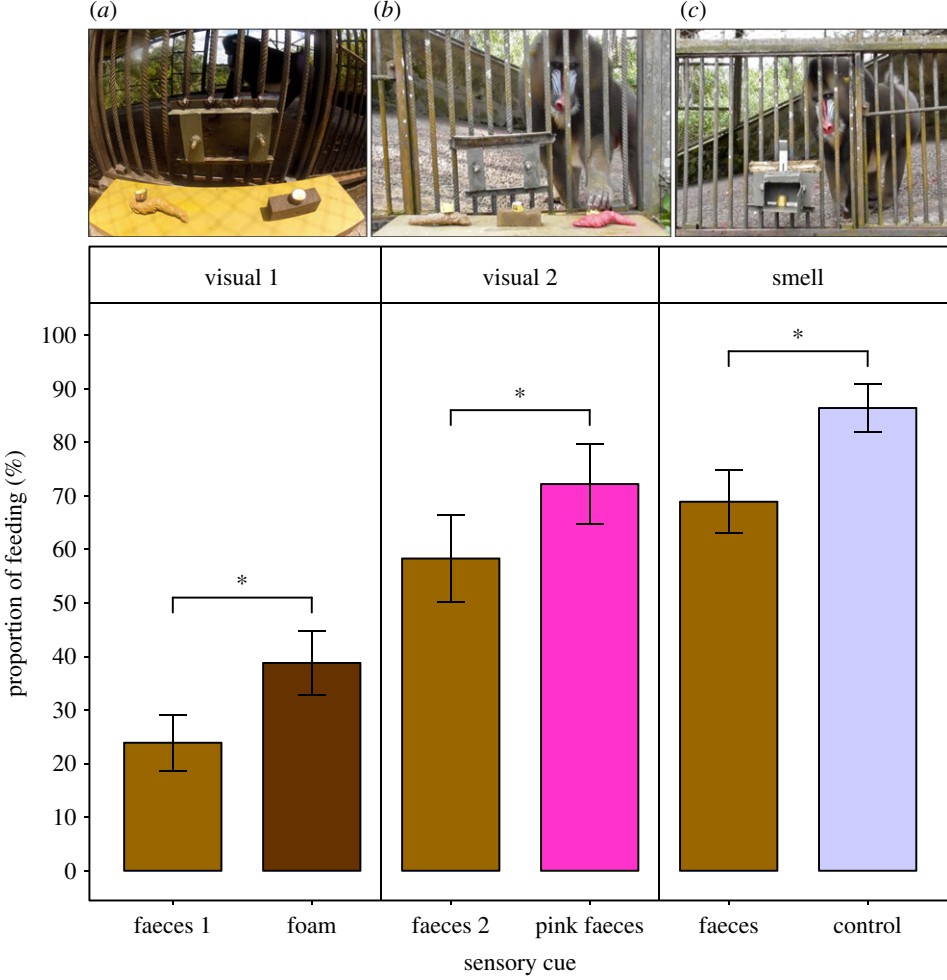

**Figure 1.** Vision- and olfaction-mediated avoidance of faeces in mandrills. (From left to right) Experimental setting to test vision-mediated avoidance of faeces in (*a*) condition 1 with brown faeces replica on the left and foam control on the right; (*b*) condition 2 with brown faeces replica, foam and pink faeces replica and in (*c*) olfaction-mediated avoidance of faeces. The bar plots represent the associated proportion of subjects feeding under either visual or olfactory cues of faeces versus controls. Bars represent the proportion of trials during which subjects fed on banana. Error bars reflect 95% binomial proportion confidence intervals and asterisks reflect significant differences between proportions (*$p < 0.05$).

24 adolescent/adult mandrills (the 10 females and 14 males that would regularly come into the isolated feeding area; average age = 11.6 ± 5.3 years, range: 5–27; see electronic supplementary material, table S3 for details) living in two natural rainforest enclosures (E1: 6.5 ha and E2: 3.5 ha) and 20 adult male long-tailed macaques (greater than 13 years) living in six different compartments of a concrete enclosure (each 10.20 × 15.20 m). To facilitate health checks and faecal sample collection, mandrills could be isolated in small elevated enclosures attached to their rainforest enclosures and long-tailed macaques in corridors accessible from their compartments through sliding doors. Neither mandrills nor macaques have been involved in biomedical research for at least 10 years. They are fed twice daily with seasonal fruits and vegetables, as well as with a baked mixture of soya beans and wheat flour.

## 2.2. Experimental procedures

All experimental procedures were approved by the Animal Welfare and Animal Care Committee of the Kyoto University Primate Research Institute (#2016-138). Experiments were conducted as far as was possible in the morning before feeding in the isolated areas for each group. Mandrills performed the tests via small apertures in the cage between isolation enclosure 1 and isolation enclosure 2 (figure 1), through which they could pass an arm and reach for food. Long-tailed macaques performed the tests via apertures in the isolation corridor's cage door, which were also large enough (3 cm between bars)

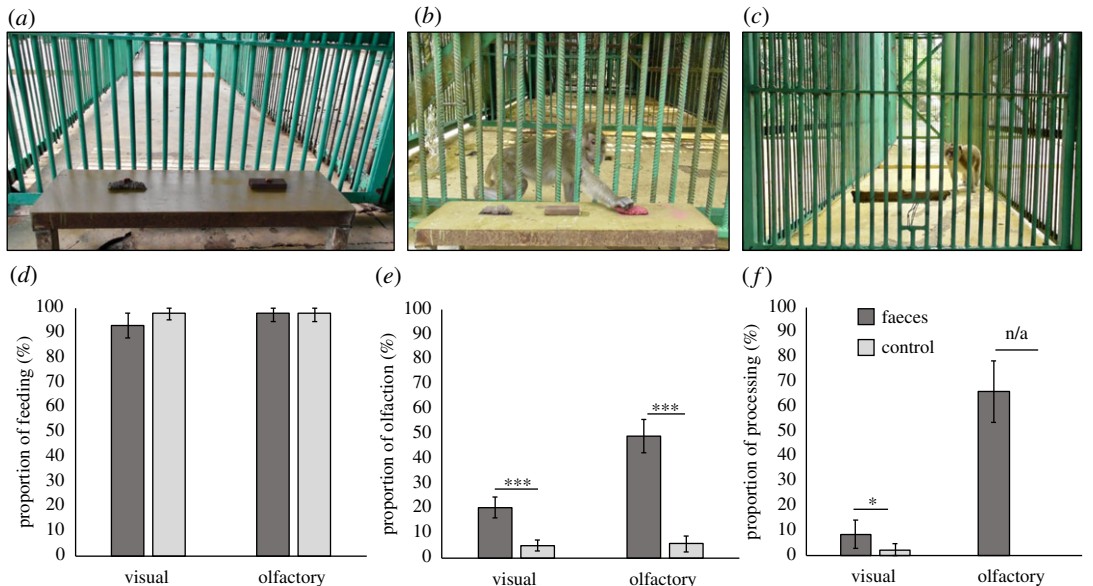

**Figure 2.** Vision- and olfaction-mediated avoidance of faeces in long-tailed macaques. (From left to right) Experimental setting to test vision- and olfaction-mediated avoidance of faeces. (*a*) Condition 1 involved brown faeces replica on the left and wood chip control on the right. (*b*) Condition 2 involved brown faeces replica, wood chip and pink faeces replica. (*c*) Photo of an olfaction-mediated avoidance of faeces experiment. Proportion of subjects (*d*) consuming food, (*e*) smelling food and (*f*) processing food associated with either visual or olfactory cues of faeces and water (control). Bars represent the proportion of trials during which subjects consumed, smelled and manipulated the food reward, respectively. Error bars reflect 95% binomial proportion confidence intervals and stars reflect significant differences between proportions (*$p < 0.05$; ***$p < 0.001$).

to allow subjects to pass an arm through and reach for food (figure 2*a,b,c*). All subjects were tested individually in the isolated areas in the sense that no other individuals could reach the experimental food, though they were not out of visual and auditory range. As a rule, the same individual was never tested twice for the same test on the same day. All experiments described below involved the presentation of food in conditions designed to test infection-risk sensitivity through visual or olfactory sensory cues. Experiments were conducted successively between October 2015 and January 2016 and recorded with a Panasonic HC-W570M or a GoPro HD Hero 2 video camera mounted on a tripod, placed 1 m away from the experimental apparatus.

## 2.3. Experiment 1: vision-mediated avoidance of faeces

In this first experiment, we presented 24 mandrills (one did not participate; i.e. did not enter the 3 m zone) and 20 long-tailed macaques with two items in condition 1: a brown piece of foam (11 × 4 cm) for mandrills or a wood chip (9 × 3 cm) for long-tailed macaques and a brown faeces replica made out of papier-mâché (15 × 6 cm for mandrills; 8 × 2.5 cm for long-tailed macaques) placed side-by-side on a metallic table (65 × 25 × 20 cm). In condition 2, the same individuals were presented with three items: the same two items as in condition 1 and a pink faeces replica, all aligned 10 cm apart on the table. All substrates were fixed on the table with double-sided tape. On each substrate, we placed the favorite food of each species (according to the animal care staff), which was either a 3 cm thick slice of banana with its skin still intact (to facilitate manipulation) for mandrills or a peanut for long-tailed macaques. Once the subject was isolated, we initiated the trial by moving the table towards the cage, within arm's reach of the subject. The test started once the subject was within 3 m of the cage bars and ended after 2.5 min. Each mandrill had 2.9 ± 0.1 trials in condition 1 (67 tests; two subjects could not be isolated for their third trial) and two trials in condition 2 (36 tests), and each macaque had five trials in condition 1 (100 tests) and two trials in condition 2 (40 tests). Tests in condition 2 were conducted after the completion of all trials in condition 1. The sides of each item on the table were alternated between tests to account for any side biases exhibited by our subjects. Note that, in condition 2, one long-tailed macaque did not participate and six mandrills could not be isolated and tested. These individuals were thus not used in the analyses.

## 2.4. Experiment 2: olfaction-mediated avoidance of faeces

In a second experiment, we tested for the avoidance of faeces via olfactory stimuli using the same individuals ($N_{mandrills} = 24$; $N_{macaques} = 20$) tested in experiment 1. To do this, we adapted different methods for mandrills and long-tailed macaques. For mandrills, we fixed a metallic feeding box ($14 \times 10 \times 13$ cm) with hooks on the cage bars. On top of it, we attached a 25 cm-long piece of bamboo horizontally on the bars (figure 1c). The bamboo was rubbed on its sides and back (i.e. top and bottom edges) with faecal material from a conspecific (approx. 4 g) or with water, which was used as a control stimulus. The box was closed from the front with a vertical sliding door and a piece of banana was placed in the box via a back door. Trials were initiated once the subject saw the box and the experimenter left the enclosure to let the subject investigate the olfactory stimulus. After 3.5 min, the front door of the box was opened by the experimenter, giving access to the piece of banana. For long-tailed macaques, we modified the experiment because subjects did not show interest in the food box when closed and refused to reach into it when open. Thus, a peanut was directly placed atop the bamboo (figure 2c) with no visible faecal material associated with it, and the experimenter moved away from the experimental area. The trial began once the subject came to within 3 m of the experimental apparatus and saw the experimenter placing the peanut on top of the piece of bamboo. For both species, each trial was terminated after consumption of the food or after 2.5 min if the food remained in the box. All subjects had three trials for each olfactory stimulus. A new sample of the same olfactory stimulus was added to the bamboo between each test when these were spaced by 10 min, because faeces would have dried and would no longer have a detectable odour (to the observer, at least). Note that faecal samples presented to subjects of one enclosure were collected from different individuals of the other enclosure in the morning of the experiments and kept in an icebox until the start of the experiment.

## 2.5. Statistical analyses

We built generalized linear mixed effects models to analyse the faeces avoidance data from the two experiments for both species. For experiment 1 with mandrills, we had four models for each condition, with feeding decision (consume or not), feeding preference (consume first or not) and olfactory investigation (smell or not) used as binary response variables across trials, and feeding latency (time in seconds before putting the food item into the mouth) used as a continuous response variable. For experiment 1 with long-tailed macaques, we had five models for each condition, including the same four responses as above plus a binary response variable to account for processing tendency (process or not)—which was only observed in macaques. For experiment 2 with mandrills, the three sets of models included feeding decision (consume or not) and olfactory investigation (smell or not) across trials as binary response variables, and feeding latency (in seconds) as a continuous variable. For experiment 2 with long-tailed macaques, we had the same three responses as with mandrills but also added a fourth set of models with processing tendency (process or not) as a binary response variable. For all models, we included substrate (control foam, brown faeces replica or pink faeces replica) or olfactory stimulus (faeces or water), dominance rank (only for macaques), age and sex (only for mandrills) and trial number as predictor variables. Random effects included individual identity (because subjects could interact with several items in each trial and these interactions were recorded as separate but dependent events) nested within group origin (to account for potential group-level variation) and trial date (as the same experiment may have been conducted on different days). Models were specified with a binomial error structure, or a negative binomial distribution for feeding latency, and logit link function. All data were analysed in R v. 3.3.3. GLMMs were fit in the packages lme4 [42] and glmmADMB [43] using maximum-likelihood estimation. For all GLMMs, we first compared the full model including the interaction between substrate or olfactory stimulus and trial number with a full model without the interaction. We retained the model with the interaction if it outperformed the model without it, determined via likelihood ratio tests (LRT) using the package lmtest [44]. The retained model was then tested against a null model that contained only an intercept term. We only present parameter estimates for the fitted models (with or without the interaction term) that significantly outperformed their respective nulls (see electronic supplementary material, table S1 and S2 for details). All raw data used in these analyses, videos of our experiments, statistical tables and tables with details of participating subjects for each experiment appear in the electronic supplementary material.

# 3. Results

## 3.1. Experiment 1: vision-mediated avoidance of faeces

### 3.1.1. Mandrills

Mandrills consumed bananas atop the brown faeces replica significantly less often than bananas atop the control foam in condition 1 (24% versus 39%, respectively; GLMM; faeces: $z = -2.48$, $p = 0.013$; table 1, figure 1$a$) and tended to increase their consumption from any substrate across trials ($z = 1.79$, $p = 0.074$). Moreover, when they fed atop both substrates, bananas atop the brown faeces replica were ingested more slowly than those atop the control foam (GLMM; faeces: $z = 2.22$, $p = 0.027$; table 1, figure 1$b$). Consumption latency across substrates decreased with age ($z = -0.72$, $p = 0.013$) and was higher in males than females ($z = 2.47$, $p = 1.1 \times 10^{-4}$). In condition 2, subjects reduced their consumption of bananas atop the brown faeces replica compared to the pink faeces replica (GLMM; brown faeces: $z = -2.07$, $p = 0.039$; table 1). However, as trial number increased (i.e. with experience), subjects increased their consumption from any substrate ($z = 2.14$, $p = 0.038$) and engaged in fewer olfactory investigations (GLMM; $z = -2.45$, $p = 0.014$; table 1).

### 3.1.2. Macaques

Long-tailed macaques did not show significant variation in their decisions to consume peanuts atop the faeces replica and peanuts atop the control wood chip in condition 1 (GLMM; faeces: $z = -1.56$, $p = 0.118$; table 2; figure 2$d$; electronic supplementary material, video S1). In addition, they increased their tendency to feed from any substrate across trials ($z = 2.20$, $p = 0.028$) and their consumption latency across substrates did not vary significantly (GLMM; faeces: $z = 1.71$, $p = 0.088$; table 2). However, they conducted significantly more olfactory investigations of peanuts placed atop faeces replicas compared to peanuts placed atop control items (GLMM; faeces: $z = 3.54$, $p = 4 \times 10^{-4}$; table 2; figure 2$e$). Subjects also decreased their tendencies to perform olfactory investigations of peanuts across trials ($z = -3.41$, $p = 6.5 \times 10^{-4}$). Macaques manipulated (rubbed, rolled) peanuts atop the faeces replica significantly more often than those atop the control (GLMM; faeces: $z = 2.38$, $p = 0.018$; table 2; figure 2$f$). In condition 2, subjects did not vary their proportion of feeding atop the pink faeces replica and the brown faeces replica (94% versus 92%, respectively; electronic supplementary material, video S2), nor their proportion of olfactory investigations (3% versus 9%) or their tendency to manipulate peanuts before ingestion (3% of all tests for both substrates).

## 3.2. Experiment 2: olfaction-mediated avoidance of faeces

### 3.2.1. Mandrills

Mandrills consumed bananas in the vicinity of faecal odour significantly less often than bananas in the vicinity of the water control (69% versus 86%; GLMM; faeces: $z = -2.53$, $p = 0.011$; table 3, figure 1$c$). We did not observe any significant difference in the proportion of olfactory investigations of bananas associated with faecal odour and those associated with water (GLMM; faeces: $z = 1.42$, $p = 0.157$; table 3). However, as trial number progressed, subjects significantly increased their likelihood to feed ($z = 3.54$, $p = 4 \times 10^{-4}$) and decreased their likelihood to smell the food before consuming it ($z = -3.05$, $p = 0.002$).

### 3.2.2. Macaques

Long-tailed macaques did not vary in their decisions to consume peanuts associated with odours of faeces or the water control (proportion of feeding across tests = 98% for both odours; figure 2$d$). However, they conducted more olfactory investigations of peanuts associated with faecal odour than in the control condition (GLMM; faeces: $z = 3.38$, $p = 7.1 \times 10^{-4}$; table 4; figure 2$e$; electronic supplementary material, video S3). In addition, although statistical models could not be performed because food manipulation was never observed in the control condition, subjects rubbed and rolled peanuts associated with faecal odour in 66% of the trials (figure 2$f$), suggesting a clear effect of the test stimulus on food manipulation behaviour. Ultimately, this resulted in a higher latency before consumption for peanuts associated with faeces odour compared to the water control (GLMM; $z = 5.66$, $p = 1.5 \times 10^{-8}$; table 4).

**Table 1.** Factors affecting variation in avoidance of visual stimuli of faeces in mandrills (experiment 1) from generalized linear mixed effects models. Italic text denotes predictor variables causing significant variation in the response: $^*p < 0.05$; $^{***}p < 0.001$.

| statistical model | predictor variable | est. | s.e. | stat. | p-value |
|---|---|---|---|---|---|
| likelihood of consumption (cond. 1) | (intercept) | −1.570 | 2.549 | −0.616 | 0.538 |
| | *item (brown faeces versus control)* | −1.689 | 0.681 | −2.482 | *0.013** |
| | age | −0.071 | 0.164 | −0.432 | 0.666 |
| | sex (males versus females) | −1.149 | 1.840 | −0.625 | 0.532 |
| | trial | 0.766 | 0.429 | 1.786 | 0.074 |
| consumption latency (cond. 1) | (intercept) | 1.828 | 0.815 | 2.24 | 0.025 |
| | *item (brown faeces versus control)* | 0.597 | 0.269 | 2.22 | *0.027** |
| | *age* | 0.085 | 0.035 | −0.72 | *0.013** |
| | *sex (males versus females)* | 1.095 | 0.283 | 2.47 | *$1.1 \times 10^{-4}$*** |
| | trial | −0.230 | 0.318 | 3.88 | 0.469 |
| likelihood of consumption (cond. 2) | (intercept) | 19.22 | 7.757 | 2.478 | 0.013 |
| | item (pink faeces versus control) | 1.857 | 1.508 | 1.232 | 0.218 |
| | *item (brown faeces versus pink faeces)* | −5.688 | 2.751 | −2.068 | *0.039** |
| | age | −0.024 | 0.293 | −0.080 | 0.936 |
| | sex (males versus females) | 0.367 | 3.741 | 0.098 | 0.922 |
| | *trial* | −3.537 | 1.706 | −2.074 | *0.038** |
| likelihood of olfactory inspection (cond. 2) | (intercept) | 7.648 | 3.573 | 2.141 | 0.032 |
| | item (pink faeces versus control) | −1.003 | 0.918 | −1.092 | 0.275 |
| | item (brown faeces versus pink faeces) | 0.282 | 0.961 | 0.294 | 0.769 |
| | age | −0.385 | 0.219 | −1.756 | 0.079 |
| | sex (males versus females) | −0.216 | 1.627 | −0.133 | 0.894 |
| | *trial* | −2.228 | 0.910 | −2.448 | *0.014** |

## 4. Discussion

Visual and olfactory cues of conspecific faeces elicited lower food consumption in mandrills and higher food manipulation (rubbing and rolling peanuts) in long-tailed macaques. Both results support the infection-avoidance hypothesis and confirm similar tendencies observed in other primate species [14,15,17,18,45]. However, mandrills expressed higher levels of aversion than long-tailed macaques, as the latter rather manipulated and investigated the contaminated food through olfaction before ingesting it. Similarly, chimpanzees tested with the same experimental paradigm did not show significant variation in feeding on control and test items, but instead prioritized the uncontaminated food for consumption [14]. That we observed only two cases of food manipulation in mandrills (using contaminated bananas) across all tests suggests that different strategies may underlie their divergent behavioural responses in comparison with the two other cercopithecoid species that have been tested (*M. fuscata* and *M. fascicularis* [18]; this study).

The tendency towards food manipulation observed in macaques may reflect a trade-off between the acquisition of energy and nutrients on the one hand and the risk of infection on the other. Peanuts represent a special treat for long-tailed macaques housed at the CIRMF and may thus elicit higher

**Table 2.** Factors affecting variation in avoidance of visual stimuli of faeces in long-tailed macaques (experiment 1) from generalized linear mixed effects models. Italic text denotes predictor variables causing significant variation in the response: $^*p < 0.05$; $^{***}p < 0.001$. Note that fitted models for condition 2 could not be performed or did not outperform their respective nulls (see electronic supplementary material, table S2).

| statistical model | predictor variable | est. | s.e. | stat. | p-value |
|---|---|---|---|---|---|
| likelihood of consumption (cond. 1) | (intercept) | 5.184 | 5.090 | 1.018 | 0.308 |
| | item (brown faeces versus control) | −1.992 | 1.275 | −1.562 | 0.118 |
| | dominance rank (low versus high) | −0.998 | 2.265 | −0.441 | 0.660 |
| | *trial* | 1.272 | 0.580 | 2.195 | *0.028** |
| consumption latency (cond. 1) | (intercept) | 2.111 | 0.508 | 4.15 | $3.3 \times 10^{-5}$ |
| | item (brown faeces versus control) | 0.244 | 0.143 | 1.71 | 0.088 |
| | dominance rank (low versus high) | 0.816 | 0.441 | 1.85 | 0.064 |
| | trial | −0.161 | 0.102 | −1.58 | 0.114 |
| likelihood of olfactory inspection of food (cond. 1) | (intercept) | −1.835 | 1.532 | −1.198 | 0.231 |
| | *item (brown faeces versus control)* | 2.944 | 0.831 | 3.542 | $4 \times 10^{-4}$*** |
| | dominance rank (low versus high) | −1.101 | 1.577 | −0.698 | 0.485 |
| | *trial* | −0.972 | 0.285 | −3.408 | $6.5 \times 10^{-4}$*** |
| likelihood of manipulating food (cond. 1) | (intercept) | −7.592 | 3.941 | −1.926 | 0.054 |
| | *item (brown faeces versus control)* | 2.950 | 1.241 | 2.376 | *0.018** |
| | dominance rank (low versus high) | −1.292 | 2.566 | −0.503 | 0.615 |
| | trial | −0.181 | 0.327 | −0.555 | 0.579 |

**Table 3.** Factors affecting variation in avoidance of olfactory stimuli of faeces in mandrills (experiment 2) from generalized linear mixed effects models. Italic text denotes predictor variables causing significant variation in the response: $^*p < 0.05$; $^{**}p < 0.01$; $^{***}p < 0.001$.

| statistical model | predictor variable | est. | s.e. | stat. | p-value |
|---|---|---|---|---|---|
| likelihood of consumption | (intercept) | −0.855 | 1.240 | −0.690 | 0.490 |
| | *odour (faeces versus control)* | −1.610 | 0.636 | −2.534 | *0.011** |
| | age | −0.029 | 0.075 | −0.391 | 0.696 |
| | sex (males versus females) | 1.186 | 0.786 | 1.510 | 0.131 |
| | *trial* | 1.731 | 0.489 | 3.543 | $4 \times 10^{-4}$*** |
| likelihood of olfactory inspection of food | (intercept) | 1.154 | 1.053 | 1.096 | 0.273 |
| | odour (faeces versus control) | 0.673 | 0.475 | 1.417 | 0.157 |
| | age | 0.003 | 0.054 | 0.062 | 0.951 |
| | sex (males versus females) | 0.736 | 0.620 | 1.186 | 0.236 |
| | *trial* | −1.004 | 0.329 | −3.049 | *0.002*** |

motivation to feed regardless of the threat of contamination. The desirability of a food item was previously found to influence feeding decisions in Japanese macaques, with subjects being much more likely to consume contaminated peanuts than contaminated wheat [18]. Modulating the value of the food item may thus have resulted in more variable feeding decisions in that study. Note, however, that using different food items, even though different items were preferred by the two species tested, might have influenced the probability of manipulation observed here; e.g. sliced bananas, even with their skins intact, may simply be less suitable for manipulation due to their softness. Whether mandrills are more likely to manipulate other food items when contaminated, or whether macaques would be less likely to manipulate items such as bananas, can only be determined through further experimentation. In addition, in the olfactory experiments, our use of

**Table 4.** Factors affecting variation in avoidance of olfactory stimuli of faeces in long-tailed macaques (experiment 2) from generalized linear mixed effects models. Italic text denotes predictor variables causing significant variation in the response: **$p < 0.01$; ***$p < 0.001$. Note that the 'likelihood of consumption' model did not outperform its respective null, and the 'likelihood of manipulating food' model was not run because subjects never processed food items in the control condition (see electronic supplementary material, table S2).

| statistical model | predictor variable | est. | s.e. | stat. | p-value |
|---|---|---|---|---|---|
| consumption latency | (intercept) | 3.208 | 0.444 | 7.23 | $4.8 \times 10^{-13}$ |
| | *odour (faeces versus control)* | 1.239 | 0.219 | 5.66 | $1.5 \times 10^{-8}$*** |
| | dominance rank (low versus high) | 0.300 | 0.351 | 0.86 | 0.392 |
| | *trial* | −0.369 | 0.139 | −2.65 | *0.008*** |
| likelihood of olfactory inspection of food | (intercept) | −3.970 | 1.605 | −2.473 | 0.013 |
| | *odour (faeces versus control)* | 4.049 | 1.197 | 3.384 | *$7.1 \times 10^{-4}$*** |
| | dominance rank (low versus high) | 0.955 | 0.900 | 1.062 | 0.288 |
| | trial | −0.501 | 0.396 | −1.264 | 0.206 |

an odourless control (water) precludes our ability to conclude that faecal odour specifically, as opposed to any other strong odour that may be aversive, affects feeding decisions. A previous study did show that the smell of a cleaning agent did not deter bonobos in the same way as faeces or rotten meat as it elicited more sensory investigations and tool uses to reach out for food [15], but future studies should use a range of odours to discriminate between competing possibilities.

Food processing in order to remove soil, sand, dust, faeces or toxins has been reported in a wide array of mammals. Carnivores such as coatis (*Nasua* spp.), skunks (*Mephitis mephitis*) and banded mongoose (*Mungos mungo*) roll millipedes on the ground with their paws to wipe off toxic secretions before eating them [46–48]. Ungulates such as boars (*Sus scrofa*) and babirusa (*Babyrousa celebensis*) use their mouths and snouts to carry soiled food items to nearby water sources and wash them [49,50]. In comparison, most primates possess an obvious anatomical advantage with fully or pseudo-opposable thumbs, which facilitates rubbing, rolling and/or washing food contaminated with soil, sand, faeces or food eliminating toxins as observed in macaques, capuchins, great apes and vervet monkeys [18,26–29,39]. Further investigation is now needed to test whether parasite avoidance could also be a driver in the evolution of food processing behaviour, as has been proposed for the removal of plant toxins and secondary compounds [51].

For both species, the aversion to contaminated food decreased across trials and across successive experiments (visual cond. 1; visual cond. 2; olfactory—only for mandrills), indicating habituation (e.g. decreased aversion). The relatively low proportions of feeding observed for mandrills in visual condition 1 of experiment 1 from atop both foam and faeces replicas may be due to the fact that both substrates were associated in all trials and separated by only ca. 25 cm. Food in the vicinity of a potential contaminant might result in some animals not feeding at all (e.g. [15]). This finding may also indicate some degree of initial neophobia towards the experiment or these substrates, as subjects had not been exposed to them prior to experimentation. Similarly, we also observed some degree of habituation while successively presenting contaminant sensory cues to chimpanzees [14] and bonobos [15]. It is likely that subjects acquired sensory information other than that targeted by this study as they were exposed to successive experiments, such as by touching faeces replicas and smelling their fingers, for example. Their tendency to sniff the food or their fingers may nonetheless vary depending on their ability to discriminate the food from the background odour. Such behaviours might also have contributed to the decreased aversion over time. Medical students who have been practising a few months of cadaver dissection reported lower disgust sensitivity in relation to touching dead, cold bodies (but not warm bodies soon after death), compared to before they started [52]. Similarly, Viar-Paxton & Olatunji [53] showed that repeated exposure to videos of a man vomiting in a bathroom reduces feelings of disgust through decreasing distress, behavioural avoidance and physiological arousal. Disgust may thus be less resistant to extinction than is fear during repeated exposure [54,55], since a potential outcome of 'predation-risk' (related to fear) is instant death, which differs fundamentally from 'parasite-risk' (related to disgust), with outcomes far less imminent. Nonetheless, habituation to disgust may vary with the level of contamination aversion sensitivity, as well as with the level of hunger in a feeding context. Subjects expressing higher contamination

aversion show slower habituation to disgust [56]. Ideally, experiments such as those we conducted with non-human primates should include a much larger sample size, avoiding repeated trials with the same individuals wherever possible to minimize habituation/sensitization effects.

In another experiment with the same groups of mandrills, subjects avoided food in contact with intact conspecific faeces [16], much more strongly than they did for food in contact with the replica faeces. However, further experiment showed that these same subjects did not appear to be averse to faecal odour when no food rewards were offered; they instead appeared attracted to the odour, spending more time in areas of the enclosure nearer its source, potentially to get information about the donor [16]. It is therefore possible that all, or at least multiple, pieces of sensory information concerning faeces are needed to elicit consistent aversion to it. Japanese macaques, for example, avoided conspecific faeces to a greater extent than they avoided faeces replicas in food-choice experiments done with free-ranging subjects [18]. Similarly, bonobos did not avoid contaminated food that was put in contact with conspecific faeces in front of them, if the latter got covered and the food alone was presented to them [15]. Finally, it is also likely that a three-way trade-off between disgust (infection risk), hunger (energy or nutrient requirements) [11,18] and information acquisition may mediate the expression of avoidance, potentially in divergent ways depending on the type of food reward and the type of contaminant or test stimulus presented, and of course the sensory modality being invoked.

As observed with chimpanzees [14], the pink faeces replica elicited an increase in feeding behaviour compared to the brown faeces replica. However, no difference in feeding strategies was observed when mandrills or chimpanzees fed atop the pink faeces replica compared to the control foam. Seeing an item with the shape of faeces but with a discordant colour seems to have relaxed their caution towards it. Similarly, in humans, seeing an image of a sanitary napkin coloured with red dye and visibly used ('pathogen-salient' cues) elicited a higher disgust score than when the sanitary napkin was coloured with a blue dye and was not used ('pathogen-free' control [57]). To our knowledge, this is the only study that tested discordant visual cues on contamination aversion, although not directly as the image described above was part of a larger set ($N = 20$) where other pathogen-salient images involved much more variants compared to the controls. Unfortunately, we only tested responses to a colour in the range of wavelengths already used in socio-sexual signalling in both species (pink), so future experiments using more neutral colours may find differing results.

In mandrills, we did not observe any sex effect on feeding decisions, despite the fact that females expressed more reluctance than males to feed on food items associated with whole conspecific faeces [16]. When presented with whole conspecific faeces, we also observed a female-bias towards caution in Japanese macaques [18]. Similarly, female grey mouse lemurs (*Microcebus murinus*) exhibited higher avoidance of food contaminated with conspecific faeces than males [17]. In bonobos, results were more challenging to interpret, because females were more risk-prone than males towards food items in contact with faeces but were more risk-averse regarding food a few centimetres away from it [15]. Therefore, more studies are needed to elucidate the mechanisms underlying potential sex differences in faeces and infection-risk aversion in non-human primates. What is clear, however, is that males tend to experience greater degrees of infection and infectious disease than females [58], so behavioural mechanisms such as differences in aversion to potential sources of infection may contribute to such explanations. Whether females who exhibit higher disgust and/or infection-risk sensitivity than males have lower levels of infection still needs further experimental investigation. We previously showed that females exhibiting more caution towards potential contaminants/contaminated items had lower levels of gastrointestinal helminth infection in Japanese macaques [18]. A recent study [17] revealed a correlation between the 'hygienic personality' of grey mouse lemurs—more pronounced in females than in males—and their gastrointestinal parasite richness. We are currently working towards further testing for such correlations with other species, e.g. mandrills, long-tailed macaques, chimpanzees and bonobos, and we encourage future work towards this end in a broader range of species under natural conditions as well.

To conclude, these findings, along with previous experiments conducted with other primate species [14,15,17–19], are consistent with the parasite avoidance theory of disgust, which proposes that aversion towards faeces and other bodily fluids, along with rotten food and other categories of disgust elicitors, have evolved to defend hosts against parasites potentially present in such contaminants [30,59]. Future studies should test whether the behavioural immune system (e.g. avoidance of contaminated food) of these animals correlates with health parameters, as was found previously for geohelminth infection in free-living macaques [18], or markers of the physiological immune system [60]. Such studies would contribute to our understanding of how avoidance behaviour fits within the evolutionary framework of parasite handling.

Ethics. Experimental procedures were approved by the Animal Welfare and Animal Care Committee of the Kyoto University Primate Research Institute (#2016-138). Research permissions were granted by the Centre International de Recherches Médicales de Franceville (CIRMF).

Data accessibility. Supporting data as well as supplementary text and videos are accessible in the electronic supplementary material, available from the Dryad Digital Repository: https://doi.org/10.5061/dryad.qz612jmb2.

Authors' contributions. C.S. and A.J.J.M. designed the experiments. B.N. facilitated the acquisition of data. C.S. conducted the experiments and analysed the data. C.S. and A.J.J.M. wrote the paper. All authors reviewed and approved the manuscript.

Competing interests. The authors declare no competing or financial interests.

Funding. C.S. was supported by the Leading Graduate Program in Primatology and Wildlife Science of Kyoto University as well as the Ministry of Education, Culture, Sports, Science and Technology of Japan (MEXT) and the Japanese Society for the Promotion of Science (JSPS).

Acknowledgements. We thank Cyr Moussadji and the staff at the CIRMF for their help conducting the experiments and members of the Social Systems and Evolution section at KUPRI for their comments. We would also like to thank three anonymous reviewers for their help in improving our manuscript.

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
