## [Reviewer comments · Royal Society Open Science]

Review History

RSOS-191861.R0 (Original submission)

Review form: Reviewer 1

Is the manuscript scientifically sound in its present form?

No

Are the interpretations and conclusions justified by the results?

No

Is the language acceptable?

Yes

Do you have any ethical concerns with this paper?

No

Have you any concerns about statistical analyses in this paper?

No

Recommendation?

Accepts with minor revision (please list in comments)

Comments to the Author(s)

An interesting study overall, but I think more clarity is needed in how the predictions were derived and how the methods were conceptualized. A more solid rationale in these areas will also aid the discussion. I elaborate below:

lines 95 - 98: how did you come to predict higher tendencies to manipulate as one of your behavioral variables? honestly, could this have come up only after the experiments were conducted, and after having seen the animals' responses? if so, that is not the way science should be done... if I have wrongly accused you of this, and there is some literature supporting manipulation as an avoidance strategy, then do cite these to justify your predictions. Otherwise, it would be better just to say that you predict animals will be more cautious, and aim to describe their avoidance strategies

line 158 onwards (description of Expt 2): It's not clear why you needed two different experimental set ups for each species. Why did you have to first hide the food from the mandrills but not the macaques? Is it because all mandrills would consume the food without fully smelling/exploring the experimental set up? If so, that is important to know, since it would mean that they may not have strong sensitivity to olfactory cues / aversion to feces odours, which would call into question their avoidance strategies under more natural conditions. And why were different food items used for each species? I'm thinking bananas might be less likely to elicit manipulation, since they would be squished and probably get even dirtier if rubbed on the ground, unlike peanuts which are dry. Could this have influenced the different avoidance responses between the two species?

lines 280-282: rather than manipulation and consumption being evidence against aversion, could the ability to process food be interpreted as an adaptation to the trade offs mentioned in lines 290-292? perhaps parasite avoidance could be a selective pressure on food processing / manipulative propensities?

Review form: Reviewer 2

Is the manuscript scientifically sound in its present form?

Yes

Are the interpretations and conclusions justified by the results?

Yes

Is the language acceptable?

Yes

Do you have any ethical concerns with this paper?

No

Have you any concerns about statistical analyses in this paper?

No

Recommendation?

Major revision is needed (please make suggestions in comments)

Comments to the Author(s)

The submitted report details an experimental test of feces avoidance by 2 primate species finding that both exhibited greater reluctance to feed on items adjacent to visual or olfactory cues of feces,

though each species responded a bit differently (one avoided, the other just gave greater inspection of the items before consuming). These data add an interesting datapoint to an existing literature on parasite avoidance.

Overall, the report is clearly written with a straightforward presentation of the study design and conclusions. The results of the experiments provide clear support for the hypothesis of feces avoidance, though the study design featured some puzzling choices that limit the interpretations. (1) I found it odd that the first experiment appeared to have been set up as a forced choice task but was not analyzed in that way. Instead of choosing the food on the right or the left, the monkeys could interact with or consume both or all three, and the analysis therefore only indirectly addresses preference. I suggest addressing which item was consumed first and/or the latency to consume each item. Given some apparent habituation to the stimuli, it is feasible this might even occur within trials, such that an initial response might be far different from a response after 2.5 minutes. (2) On the other hand, given that the analysis is merely concerned with whether each item was consumed/inspected, they might better have been presented individually, because the experiment leaves open the potential problem that the presence of a feces replica on the table might have affected both choices. For example, having worked with captive primates, I find it extraordinary that the mandrills consumed less than half of the bananas associated with the control foam...could this be because the feces replica was still pretty close? (3) While it was clear that the macaques tendency to rub or smell the items was much different depending on the associated stimulus, and that clearly supports the hypothesis, you should be more cautious about the conclusion that this is a species difference in strategy, as the nature of the food items differed. It isn't very realistic to rub a banana slice around the floor, and the rolling of the peanut has a potential practical purpose in that it may remove the skin. Even the tendency to sniff might vary depending on how well the food item can be discriminated against a background odor. (4) To that end, the control condition in the olfactory experiment is not ideal because it is odorless. This leaves open the possibility that any strong odor might reduce consumption rates and/or lead to further inspection of the food item. (5) Finally, given the importance of pink/red hues in visual signalling in both of these species, it is unfortunate that pink was used to test for the response to discordant cues. It is quite reasonable to think that these monkeys would be more strongly attracted to pink cues of any kind.

Some of these issues can/should be addressed with additional data or with tweaks to the data analysis, but the limitations certainly need to be more fully discussed.

My only other major critique is that, while the manuscript does a nice job of setting out why feces avoidance is an important adaptation, I do not think the introduction provides a strong enough rationale for this particular study. It is clear that feces avoidance is a well-established behavioral phenomenon, even in primates. The authors primarily point to a lack of aversion to an individual's own feces as a potential counterpoint for primates, though they themselves note that this is of limited relevance. So, what advance is this study hoping to make other than adding a new datapoint? In particular, as the study design involves testing of alternative discrimination mechanisms and a comparison between two species, did they authors have any theoretically informed predictions about the differences that could contribute to the broader theory?

Decision letter (RSOS-191861.R0)

27-Nov-2019

Dear Ms Sarabian,

The editors assigned to your paper ("Divergent strategies in feces avoidance between two cercopithecoid primates") have now received comments from reviewers. We would like you to

revise your paper in accordance with the referee and Associate Editor suggestions which can be found below (not including confidential reports to the Editor). Please note this decision does not guarantee eventual acceptance.

Please submit a copy of your revised paper before 20-Dec-2019. Please note that the revision deadline will expire at 00.00am on this date. If we do not hear from you within this time then it will be assumed that the paper has been withdrawn. In exceptional circumstances, extensions may be possible if agreed with the Editorial Office in advance. We do not allow multiple rounds of revision so we urge you to make every effort to fully address all of the comments at this stage. If deemed necessary by the Editors, your manuscript will be sent back to one or more of the original reviewers for assessment. If the original reviewers are not available, we may invite new reviewers.

- Data accessibility

<http://datadryad.org/submit?journalID=RSOS&manu=RSOS-191861>

- Competing interests

- Authors' contributions

All submissions, other than those with a single author, must include an Authors' Contributions section which individually lists the specific contribution of each author. The list of Authors

should meet all of the following criteria; 1) substantial contributions to conception and design, or acquisition of data, or analysis and interpretation of data; 2) drafting the article or revising it critically for important intellectual content; and 3) final approval of the version to be published.

- Acknowledgements

- Funding statement

on behalf of Dr Atsushi Iriki (Associate Editor) and Professor Kevin Padian (Subject Editor)
openscience@royalsociety.org

Editors' Comments to Author:

Thanks for your submission. As you will see the reviewers have some reservations about the procedures in the study and also about its importance. "Adding another data point" may not be without value, and we would not want people to overstate their importance, but what is being added to the literature that is a noteworthy result is an important consideration. Please clarify this in your revision so that we can better judge the acceptability of the manuscript. Thanks so much for submitting.

Reviewers' Comments to Author:

Reviewer: 1

Comments to the Author(s)

An interesting study overall, but I think more clarity is needed in how the predictions were derived and how the methods were conceptualized. A more solid rationale in these areas will also aid the discussion. I elaborate below:

lines 95 - 98: how did you come to predict higher tendencies to manipulate as one of your behavioral variables? honestly, could this have come up only after the experiments were conducted, and after having seen the animals' responses? if so, that is not the way science should

be done... if I have wrongly accused you of this, and there is some literature supporting manipulation as an avoidance strategy, then do cite these to justify your predictions. Otherwise, it would be better just to say that you predict animals will be more cautious, and aim to describe their avoidance strategies

line 158 onwards (description of Expt 2): It's not clear why you needed two different experimental set ups for each species. Why did you have to first hide the food from the mandrills but not the macaques? Is it because all mandrills would consume the food without fully smelling/exploring the experimental set up? If so, that is important to know, since it would mean that they may not have strong sensitivity to olfactory cues / aversion to feces odours, which would call into question their avoidance strategies under more natural conditions. And why were different food items used for each species? I'm thinking bananas might be less likely to elicit manipulation, since they would be squished and probably get even dirtier if rubbed on the ground, unlike peanuts which are dry. Could this have influenced the different avoidance responses between the two species?

lines 280-282: rather than manipulation and consumption being evidence against aversion, could the ability to process food be interpreted as an adaptation to the trade offs mentioned in lines 290-292? perhaps parasite avoidance could be a selective pressure on food processing / manipulative propensities?

Reviewer: 2

Comments to the Author(s)

The submitted report details an experimental test of feces avoidance by 2 primate species finding that both exhibited greater reluctance to feed on items adjacent to visual or olfactory cues of feces, though each species responded a bit differently (one avoided, the other just gave greater inspection of the items before consuming). These data add an interesting datapoint to an existing literature on parasite avoidance.

Overall, the report is clearly written with a straightforward presentation of the study design and conclusions. The results of the experiments provide clear support for the hypothesis of feces avoidance, though the study design featured some puzzling choices that limit the interpretations. (1) I found it odd that the first experiment appeared to have been set up as a forced choice task but was not analyzed in that way. Instead of choosing the food on the right or the left, the monkeys could interact with or consume both or all three, and the analysis therefore only indirectly addresses preference. I suggest addressing which item was consumed first and/or the latency to consume each item. Given some apparent habituation to the stimuli, it is feasible this might even occur within trials, such that an initial response might be far different from a response after 2.5 minutes. (2) On the other hand, given that the analysis is merely concerned with whether each item was consumed/inspected, they might better have been presented individually, because the experiment leaves open the potential problem that the presence of a feces replica on the table might have affected both choices. For example, having worked with captive primates, I find it extraordinary that the mandrills consumed less than half of the bananas associated with the control foam...could this be because the feces replica was still pretty close? (3) While it was clear that the macaques tendency to rub or smell the items was much different depending on the associated stimulus, and that clearly supports the hypothesis, you should be more cautious about the conclusion that this is a species difference in strategy, as the nature of the food items differed. It isn't very realistic to rub a banana slice around the floor, and the rolling of the peanut has a potential practical purpose in that it may remove the skin. Even the tendency to sniff might vary depending on how well the food item can be discriminated against a background odor. (4) To that end, the control condition in the olfactory experiment is not ideal because it is odorless. This leaves open the possibility that any strong odor might reduce consumption rates and/or lead to further inspection of the food item. (5) Finally, given the importance of pink/red hues in visual signalling in both of these species, it is unfortunate that pink was used to test for the response to

discordant cues. It is quite reasonable to think that these monkeys would be more strongly attracted to pink cues of any kind.

Some of these issues can/should be addressed with additional data or with tweaks to the data analysis, but the limitations certainly need to be more fully discussed.

My only other major critique is that, while the manuscript does a nice job of setting out why feces avoidance is an important adaptation, I do not think the introduction provides a strong enough rationale for this particular study. It is clear that feces avoidance is a well-established behavioral phenomenon, even in primates. The authors primarily point to a lack of aversion to an individual's own feces as a potential counterpoint for primates, though they themselves note that this is of limited relevance. So, what advance is this study hoping to make other than adding a new datapoint? In particular, as the study design involves testing of alternative discrimination mechanisms and a comparison between two species, did they authors have any theoretically informed predictions about the differences that could contribute to the broader theory?

Author's Response to Decision Letter for (RSOS-191861.R0)

See Appendix A.

RSOS-191861.R1 (Revision)

Review form: Reviewer 3

Is the manuscript scientifically sound in its present form?

Yes

Are the interpretations and conclusions justified by the results?

Yes

Is the language acceptable?

Yes

Do you have any ethical concerns with this paper?

No

Have you any concerns about statistical analyses in this paper?

No

Recommendation?

Accept with minor revision (please list in comments)

Comments to the Author(s)

Review to Royal Society Open Science

I think the authors have made a good job reviewing the manuscript in the light of the referees' comments. There is now more rationale for why they investigated behavioural responses to visual and olfactory stimuli of faeces, more detail about method choices, additional analysis, and a stronger discussion regarding potential limitations associated to the experimental design. Although the report has substantial merit and can contribute to the literature on parasites

avoidance, I think further revision is need before it can be considering for publication. Below I detail the issues that still require appropriate resolution.

The introduction is still lacking theoretical rationale and a clear prediction about the comparison between the two primate species (long-tailed macaques and mandrills). How it can contribute to the broader literature and what is expected (differences or not and why)? These are even more required considering the current write-up, that presents this species comparison as one out the study's aims.

Line 75: Replace "Cebus" with "Sapajus" according to Lima et al 2017, *Mol. Phylogenet. Evol.*, 124, 137-150.

Lines 90-91: It sounds like a post hoc goal, since no theoretical basement nor prediction were provided in order to compare the two species.

Lines 104-105: I wonder how the researchers selected the subset of individuals whose participated in the experiments from out approximately 200 mandrills available.

Line 203: As described, "olfactory investigation" is also a binary response variable.

Discussion

Lines 293-294: It contradicts the introduction, where primates were described as "liberal when it comes to their disposition towards faeces". I suggest reformulating the argument and provide the relevant references that support the infection avoidance hypothesis in primates.

Overall, the major criticism I have is that the experimental design differs from one tested species to another, which may account for part of the results (inter-specific differences) and limit the interpretations. Even though, I recognize that in the current version the authors addressed the study limitations and proposed suggestions for further investigation on this field.

Decision letter (RSOS-191861.R1)

14-Feb-2020

Dear Ms Sarabian,

On behalf of the Editors, I am pleased to inform you that your Manuscript RSOS-191861.R1 entitled "Divergent strategies in faeces avoidance between two cercopithecoid primates" has been accepted for publication in Royal Society Open Science subject to minor revision in accordance with the referee suggestions. Please find the referees' comments at the end of this email.

The reviewers and Subject Editor have recommended publication, but also suggest some minor revisions to your manuscript. Therefore, I invite you to respond to the comments and revise your manuscript.

- Ethics statement

- Data accessibility

It is a condition of publication that all supporting data are made available either as supplementary information or preferably in a suitable permanent repository. The data accessibility section should state where the article's supporting data can be accessed. This section should also include details, where possible of where to access other relevant research materials

such as statistical tools, protocols, software etc can be accessed. If the data has been deposited in an external repository this section should list the database, accession number and link to the DOI for all data from the article that has been made publicly available. Data sets that have been deposited in an external repository and have a DOI should also be appropriately cited in the manuscript and included in the reference list.

If you wish to submit your supporting data or code to Dryad (<http://datadryad.org/>), or modify your current submission to dryad, please use the following link:
<http://datadryad.org/submit?journalID=RSOS&manu=RSOS-191861.R1>

- **Competing interests**

- **Authors' contributions**

- **Acknowledgements**

- **Funding statement**

Because the schedule for publication is very tight, it is a condition of publication that you submit the revised version of your manuscript before 23-Feb-2020. Please note that the revision deadline will expire at 00.00am on this date. If you do not think you will be able to meet this date please let me know immediately.

When submitting your revised manuscript, you will be able to respond to the comments made by the referees and upload a file "Response to Referees" in "Section 6 - File Upload". You can use this

to document any changes you make to the original manuscript. In order to expedite the processing of the revised manuscript, please be as specific as possible in your response to the referees.

on behalf of Dr Atsushi Iriki (Associate Editor) and Kevin Padian (Subject Editor)
openscience@royalsociety.org

Reviewer comments to Author:

Reviewer: 3
Comments to the Author(s)

Review to Royal Society Open Science

I think the authors have made a good job reviewing the manuscript in the light of the referees' comments. There is now more rationale for why they investigated behavioural responses to visual and olfactory stimuli of faeces, more detail about method choices, additional analysis, and a stronger discussion regarding potential limitations associated to the experimental design. Although the report has substantial merit and can contribute to the literature on parasitic avoidance, I think further revision is needed before it can be considered for publication. Below I detail the issues that still require appropriate resolution.

The introduction is still lacking theoretical rationale and a clear prediction about the comparison between the two primate species (long-tailed macaques and mandrills). How it can contribute to the broader literature and what is expected (differences or not and why)? These are even more required considering the current write-up, that presents this species comparison as one out the study's aims.

Line 75: Replace "Cebus" with "Sapajus" according to Lima et al 2017, Mol. Phylogenet. Evol., 124, 137-150.

Lines 90-91: It sounds like a post hoc goal, since no theoretical basement nor prediction were provided in order to compare the two species.

Lines 104-105: I wonder how the researchers selected the subset of individuals whose participated in the experiments from out approximately 200 mandrills available.

Line 203: As described, "olfactory investigation" is also a binary response variable.

Discussion

Lines 293-294: It contradicts the introduction, where primates were described as "liberal when it comes to their disposition towards faeces". I suggest reformulating the argument and provide the relevant references that support the infection avoidance hypothesis in primates.

Overall, the major criticism I have is that the experimental design differs from one tested species to another, which may account for part of the results (inter-specific differences) and limit the interpretations. Even though, I recognize that in the current version the authors addressed the study limitations and proposed suggestions for further investigation on this field.

Author's Response to Decision Letter for (RSOS-191861.R1)

See Appendix B.

Decision letter (RSOS-191861.R2)

24-Feb-2020

Dear Ms Sarabian,

It is a pleasure to accept your manuscript entitled "Divergent strategies in faeces avoidance between two cercopithecoïd primates" in its current form for publication in Royal Society Open Science. The comments of the reviewer(s) who reviewed your manuscript are included at the foot of this letter.

Please ensure that you send to the editorial office individual files for each figure and table included in your manuscript.

You can send these in a zip folder if more convenient. Failure to provide these files may delay the processing of your proof. You may disregard this request if you have already provided these files to the editorial office.

You can expect to receive a proof of your article in the near future. Please contact the editorial office (openscience_proofs@royalsociety.org) and the production office (openscience@royalsociety.org) to let us know if you are likely to be away from e-mail contact -- if

you are going to be away, please nominate a co-author (if available) to manage the proofing process, and ensure they are copied into your email to the journal.

on behalf of Dr Atsushi Iriki (Associate Editor) and Kevin Padian (Subject Editor)
openscience@royalsociety.org

Appendix A

Editorial Board
Royal Society Open Science

December 21, 2019

Dear Dr. Atsushi Iriki and Professor Kevin Padian,

We thank you and the reviewers for the constructive comments on our earlier submission of the manuscript, “Divergent strategies in faeces avoidance between two cercopithecoid primates” for consideration in *Royal Society Open Science*.

In response to the reviewers’ comments, we have made several changes to the revised version of the manuscript. To briefly summarise the changes we have made, in response to Reviewer 1’s comments, we clarified how our predictions regarding faeces avoidance behaviours were derived and how our methods were conceptualized. In response to additional comments from Reviewer 2, we conducted further analyses to provide more information regarding feeding behaviour in our faeces avoidance experiments (-feeding latency is higher for food associated with visual and olfactory cues of faeces when models worked), stipulated potential limitations associated to our method choices, and tried to emphasize throughout the text our rationale for conducting this study.

Our responses to the reviewers’ comments are inserted after each comment below in blue text.

We thank you for your time considering our manuscript and we look forward to hearing back from you in the near future.

Sincerely,

Cécile Sarabian, D. Sc.
Postdoctoral fellow of the Japan Society for the Promotion of Science
Kyoto University Primate Research Institute
41-2 Kanrin, Inuyama, Aichi
484-8506 Japan
sarabiancecile@gmail.com

DETAILED RESPONSES TO REVIEWERS' COMMENTS:

Reviewer: 1

Comments to the Author(s)

An interesting study overall, but I think more clarity is needed in how the predictions were derived and how the methods were conceptualized. A more solid rationale in these areas will also aid the discussion.

We thank the reviewer for this assessment of our work and revised our manuscript based on these suggestions. Please, see below.

I elaborate below:

lines 95 - 98: how did you come to predict higher tendencies to manipulate as one of your behavioral variables? honestly, could this have come up only after the experiments were conducted, and after having seen the animals' responses? if so, that is not the way science should be done... if I have wrongly accused you of this, and there is some literature supporting manipulation as an avoidance strategy, then do cite these to justify your predictions. Otherwise, it would be better just to say that you predict animals will be more cautious, and aim to describe their avoidance strategies.

Long-tailed macaques, Japanese macaques, capuchins, vervets and great apes have previously been observed to manipulate/wash food contaminated with soil or sand (Visalberghi & Fragaszy 1990, *Anim. Behav.*; Sarabian & MacIntosh 2015, *Biol. Lett.*; Van de Waal et al. 2014, *Anim. Behav.*; Allritz et al. 2013, *Primates*; Neadle et al. 2018, *PLOS One*). Similarly, northern pig-tailed macaques and Japanese macaques on Koshima islet would process toxic caterpillars and even earthworms before eating them (Trébouet et al. 2018, *Primates*; Sarabian *pers. obs.*). Regarding feces contamination, our previous observations show that Japanese macaques would rub grains of wheat and peanuts that came from the top of conspecific feces or feces replica before ingestion, and they commonly manipulate foods picked up off the ground before ingestion (Sarabian & MacIntosh, *Biol. Lett.*; *pers. obs.*). Finally, long-tailed macaques at the CIRMF were also observed to rub/manipulate food during pilot observations prior to experiments. We thus added some rationale based on this literature for our prediction regarding food manipulation L72-77.

line 158 onwards (description of Expt 2): It's not clear why you needed two different experimental set ups for each species. Why did you have to first hide the food from the mandrills but not the macaques? Is it because all mandrills would consume the food without fully smelling/exploring the experimental set up? If so, that is important to know, since it would mean that they may not have strong sensitivity to olfactory cues / aversion to feces odours, which would call into question their avoidance strategies under more natural conditions. And why were different food items used for each species? I'm thinking bananas might be less likely to elicit manipulation, since they would be squished and probably get even dirtier if rubbed on the ground, unlike peanuts which are dry. Could this have influenced the different avoidance responses between the two species?

We agree that not having parallel experiments makes interpretation somewhat unclear. When we tried the 'olfactory box' with the macaques (a smaller adapted version of the box used for mandrills), subjects would not put their hands into the box to get the food. As such, we adapted the experimental design to the macaques by putting the food directly on top of the piece of bamboo containing the olfactory cue. We now reflect on this in the Methods L175-177. According to perceptions of the animal care staff, the preferred food items for each species differed, and this was reflected in our experiments: we used bananas for mandrills and peanuts for macaques. We added this information in the Methods L148-150

and we discuss how this difference in food preference and presentation may have influenced behavioral responses L313-316.

lines 280-282: rather than manipulation and consumption being evidence against aversion, could the ability to process food be interpreted as an adaptation to the trade-offs mentioned in lines 290-292? perhaps parasite avoidance could be a selective pressure on food processing / manipulative propensities?

As mentioned lines 305-307, the tendency towards food manipulation may be an adaptation to the infection-risk vs. energy requirement trade-off. We thought of the hypothesis that parasite avoidance might be a selective pressure promoting food processing, although this would need further investigation. We nonetheless suggest it L327-339, based on previous findings convergent with this hypothesis.

Reviewer: 2

Comments to the Author(s)

The submitted report details an experimental test of feces avoidance by 2 primate species finding that both exhibited greater reluctance to feed on items adjacent to visual or olfactory cues of feces, though each species responded a bit differently (one avoided, the other just gave greater inspection of the items before consuming). These data add an interesting data point to an existing literature on parasite avoidance.

Overall, the report is clearly written with a straightforward presentation of the study design and conclusions. The results of the experiments provide clear support for the hypothesis of feces avoidance, though the study design featured some puzzling choices that limit the interpretations.

We thank the reviewer for his/her comments, which converge with Reviewer 1's comments. We provide more information regarding our study design choice below.

(1) I found it odd that the first experiment appeared to have been set up as a forced choice task but was not analyzed in that way. Instead of choosing the food on the right or the left, the monkeys could interact with or consume both or all three, and the analysis therefore only indirectly addresses preference. I suggest addressing which item was consumed first and/or the latency to consume each item. Given some apparent habituation to the stimuli, it is feasible this might even occur within trials, such that an initial response might be far different from a response after 2.5 minutes.

The feeding preference (response: feed first on each substrate as 1/0) was indeed previously recorded for each condition of the Exp. 1 in both species. However, models for this response did not outperform their respective null models in either mandrills or macaques (see Supplementary results Exp. 1 and Table S1). We re-analyzed the videos, this time considering the latency to feed atop each substrate, when subjects fed. Models outperformed their respective nulls in the first condition for both species but not in the second condition (see Supplementary results Exp. 1 and Table S1). Results show that mandrills took more time to feed atop the faeces replica compared to the control, and macaques tended to do the same (although the difference was not significant). We have now added these into the Statistical analyses, the Results (LL235-239, 251-252, 285-287) and in Tables 1, 2, and 4.

(2) On the other hand, given that the analysis is merely concerned with whether each item was consumed/inspected, they might better have been presented individually, because the experiment leaves open the potential problem that the presence of a feces replica on the table might have affected

both choices. For example, having worked with captive primates, I find it extraordinary that the mandrills consumed less than half of the bananas associated with the control foam...could this be because the feces replica was still pretty close?

We thank the reviewer for this comment. It's certainly true that food in the vicinity of a potential contaminant might result in some animals not feeding at all. We observed the same phenomenon in bonobos (Sarabian et al. 2018, *Phil. Trans. B*). We added this L344-346. It may have been useful to test the items independently, but this was not done. However, considering we still see significant differences in consumption probabilities for the feces replica versus the control item, we feel the existing study design still results in interpretable results. Had we had negative results, then study design may indeed have been a critical factor there. Just additionally, one alternative hypothesis regarding the fact that subjects consumed less than half of the bananas associated with the control foam – based on our experience working with mandrills – is that it might rather be linked to a higher neophobia or stress linked to the isolation than the feces replica around (see L346-348).

(3) While it was clear that the macaques tendency to rub or smell the items was much different depending on the associated stimulus, and that clearly supports the hypothesis, you should be more cautious about the conclusion that this is a species difference in strategy, as the nature of the food items differed. It isn't very realistic to rub a banana slice around the floor, and the rolling of the peanut has a potential practical purpose in that it may remove the skin. Even the tendency to sniff might vary depending on how well the food item can be discriminated against a background odor.

Right, our choice of favorite food might have been an additional constrain for mandrills to not have rubbed the food (see L313-316). However, according to previous literature and based on observations, macaques seem more inclined than mandrills to process/manipulate food (see first response to Reviewer 1). Additionally, peanut manipulation behavior has still been observed in macaques even when the skin is removed. We agree with the fact that the tendency to sniff the food may also vary depending on their ability to discriminate the food from the background odor. This is now added L353-355.

(4) To that end, the control condition in the olfactory experiment is not ideal because it is odorless. This leaves open the possibility that any strong odor might reduce consumption rates and/or lead to further inspection of the food item.

Right, this is what we realized and could add in further experiments with bonobos using detergent as another control – which did not deter them. On the contrary, it provoked more sensory investigations and tool uses to reach out for food compared to contaminant odors (see Sarabian et al. 2018, *Phil. Trans. B*). We also reflect on this L319-325.

(5) Finally, given the importance of pink/red cues in visual signaling in both of these species, it is unfortunate that pink was used to test for the response to discordant cues. It is quite reasonable to think that these monkeys would be more strongly attracted to pink cues of any kind.

This is an interesting and potentially confounding point that we had not considered but may indeed be relevant. Our choice for pink color was based simply on wanting something that clearly contrasts with the range of possible faecal colors available: e.g. green, yellow, red, orange, and black were excluded as potential variations in faecal color. However, shades of blue might have provided a better option considering the point raised. This is perhaps something we can consider in future experiments. We added a note about this in the text on L401-403.

Some of these issues can/should be addressed with additional data or with tweaks to the data analysis, but the limitations certainly need to be more fully discussed.

We have taken into consideration all the comments from the reviewer to the best of our ability, including further data analyses, and have now added some relevant information in the manuscript.

My only other major critique is that, while the manuscript does a nice job of setting out why feces avoidance is an important adaptation, I do not think the introduction provides a strong enough rationale for this particular study. It is clear that feces avoidance is a well-established behavioral phenomenon, even in primates. The authors primarily point to a lack of aversion to an individual's own feces as a potential counterpoint for primates, though they themselves note that this is of limited relevance. So, what advance is this study hoping to make other than adding a new data point? In particular, as the study design involves testing of alternative discrimination mechanisms and a comparison between two species, did the authors have any theoretically informed predictions about the differences that could contribute to the broader theory?

As stated L89-91 and L96-99, this comparative study was still rather exploratory given the few things we know regarding sensory-mediated parasite avoidance responses in non-human primates. According to R1's comments as well, we have now added a stronger rationale for why we investigated behavioural responses to visual and olfactory stimuli of feces throughout the text.

Appendix B

Editorial Board
Royal Society Open Science

February 22, 2020

Dear Dr. Atsushi Iriki and Dr. Kevin Padian,

We thank you and the reviewer for the additional constructive comments on our resubmission of the manuscript, “Divergent strategies in faeces avoidance between two cercopithecoid primates” for consideration in *Royal Society Open Science*.

In response to Reviewer 3’s comments, we have made the additional requested changes to the second revised version of the manuscript. To briefly summarise the changes we have made, we gave more rationale for the expected species difference and a clearer prediction in the Introduction, clarified a point in the methods, and added a few references to support the infection-avoidance hypothesis in other primate species in the Discussion.

Our responses to the reviewer’s comments are inserted after each comment below in blue text.

We thank you for your time considering and accepting our manuscript.

Sincerely,

Cécile Sarabian, D. Sc.
Postdoctoral fellow of the Japan Society for the Promotion of Science
Kyoto University Primate Research Institute
41-2 Kanrin, Inuyama, Aichi
484-8506 Japan
sarabiancecile@gmail.com

DETAILED RESPONSES TO REVIEWER'S COMMENTS:

Reviewer: 3

Comments to the Author(s)

I think the authors have made a good job reviewing the manuscript in the light of the referees' comments. There is now more rationale for why they investigated behavioural responses to visual and olfactory stimuli of faeces, more detail about method choices, additional analysis, and a stronger discussion regarding potential limitations associated to the experimental design. Although the report has substantial merit and can contribute to the literature on parasites avoidance, I think further revision is need before it can be considering for publication. Below, I detail the issues that still require appropriate resolution.

We thank the reviewer for this assessment of our work and revised our manuscript based on these suggestions. Please, see below.

The introduction is still lacking theoretical rationale and a clear prediction about the comparison between the two primate species (long-tailed macaques and mandrills). How it can contribute to the broader literature and what is expected (differences or not and why)? These are even more required considering the current write-up, that presents this species comparison as one out the study's aims.

We have now added more rationale and a clear prediction between the two species in our introduction, which may serve the broader literature on the potential function of food processing and the diversity of strategies regarding parasite avoidance. Please, see L91-94 and L98-99, respectively. In short, we report that several species of macaques have been previously observed to process contaminated food, whereas no report exists for mandrills, as far as we know. From this, comes the prediction that macaques would show higher tendencies to manipulate food – if the food is of enough value to them.

Line 75: Replace "Cebus" with "Sapajus" according to Lima et al 2017, Mol. Phylogenet. Evol., 124, 137-150.

Done.

Lines 90-91: It sounds like a post hoc goal, since no theoretical basement nor prediction were provided in order to compare the two species.

We have now clarified this by referring to previous literature in order to make a clearer prediction (see above).

Lines 104-105: I wonder how the researchers selected the subset of individuals whose participated in the experiments from out approximately 200 mandrills available.

We have now clarified this, L112, by stating that the selected subset of mandrills came from the easiness of isolating them (while keeping a fair sex ratio).

Line 203: As described, "olfactory investigation" is also a binary response variable. This has been added.

Discussion

Lines 293-294: It contradicts the introduction, where primates were described as "liberal when it comes

to their disposition towards faeces". I suggest reformulating the argument and provide the relevant references that support the infection avoidance hypothesis in primates.

We have now reminded in the first line of our discussion that visual and olfactory cues of faeces refer to conspecific faeces, and not their own. We have also added the references supporting the infection-avoidance hypothesis in primates, L300. We also made it clearer in the Introduction (L54-63) that the "liberality" primates may take towards faeces (1) is observed mainly in abnormal conditions of captivity and/or (2) mainly concerns their own waste products, which pose significantly fewer risks of encountering novel parasites.

Overall, the major criticism I have is that the experimental design differs from one tested species to another, which may account for part of the results (inter-specific differences) and limit the interpretations. Even though, I recognize that in the current version the authors addressed the study limitations and proposed suggestions for further investigation on this field.

Thank you for the critical review and additional comments regarding our work. We have made clear that the experimental design was adapted to each species depending on their food preference and their initial reactions to the box designed for olfactory experiments. We admit, however, that identical conditions would have made for a stronger species comparison, and must accept this as a limitation of our work.